# DISTINCT AND SHARED CONCEPT DISCOVERY FOR FINE-GRAINED CONCEPT INVERSION

## ABSTRACT

A real-world object is expressed by composing distinctive characteristics that distinguish it from others and some common properties shared with different objects. Recent advances in generative modeling focus on identifying the shared concepts within images of individual identities. However, it remains unclear how to identify shared concepts beyond multiple identities while preserving the unique concepts inherent to each. In this work, we address this new problem of simultaneously discovering similarities and differences between two sets of images and propose a two-stage framework coined `DISCOD` (DIstinct and Shared COncept Discovery). In the first stage of `DISCOD`, we introduce information-regularized textual inversion, focusing on separating representative concepts distinctive from others while capturing the shared concepts among different objects. In the next stage, we further optimize them to align composited concepts of those with the corresponding objects, respectively. We demonstrate the effectiveness of `DISCOD` by showing that `DISCOD` discovers the concepts better than baselines, as measured by CLIPScore and success rate. The human study also validates the reasonable discovery capability of `DISCOD`. Furthermore, we show the practical applicability of our approach by applying to various applications: image editing, few-shot personalization of diffusion models, and group bias mitigation in recognition.

## 1    INTRODUCTION

Concepts (Smith & Medin, 1981) are essential notions that define and describe an object and range from concrete notions of attributes like color and shape to abstract ones like functionality. Given a set of visual objects, the objects are represented by a combination of common concepts shared across them and distinct concepts for each individual object or each subset of the objects. Recognizing these shared and distinct visual concepts across objects is beneficial in various fields, including taxonomy definition, which improves our understanding of categories (Zhao et al., 2024), the creation of novel objects,[1] and the efficient learning of new concepts (Lake et al., 2015).

Despite the fundamental benefits of recognizing concepts, discovering visual concepts from images remains challenging due to factors such as complex object composition in real-world scenes and entanglement with various other concepts (Huang et al., 2023a). Recent works (Gal et al., 2023; Vinker et al., 2023; Chefer et al., 2024; Avrahami et al., 2023) have proposed concept discovery methods from the set of input images, where the images contain object instances sharing at least one concept. These methods optimize a textual concept embedding that encodes a common concept or multiple embeddings by decomposing shared sub-concepts recognizable by Vision Language Models (VLMs),[2] such as CLIP (Radford et al., 2021) or Diffusion models (Rombach et al., 2022). Although these approaches have proven useful for many applications of image synthesis (Gal et al., 2023; Safaee et al., 2024; Ruiz et al., 2023; Sohn et al., 2023; Avrahami et al., 2023; Huang et al., 2023b; Kumari et al., 2023; Liu et al., 2023a), such as personalization, editing, and compositing, and understanding sub-concepts of a given object in a human interpretable form (Gal et al., 2023; Chefer et al., 2024), these approaches primarily focus on extracting commonalities and are not capable of identifying differences.

---

[1]https://clios.com/awards
[2]We will refer to text-to-image diffusion with CLIP as VLMs.

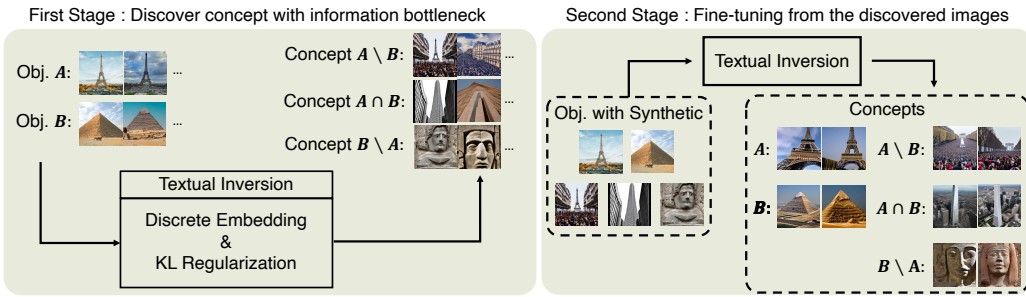

Figure 1: `DISCOD`, **the two-stage framework to discover distinct and shared concepts.** In the first stage, we discover the textual tokens of shared and distinct concepts with information-regularized textual inversion: pre-trained discrete token embedding and KL regularization. In the second stage, we generate the images using the discovered concepts from the first stage, then further optimize the textual token to align the given two objects.

In this work, we focus on a new discovery problem identifying both shared and distinct concepts within and between two groups of objects. To recognize the commonalities and differences, we need to analyze objects into disentangled concepts and then compare these objects comprehensively according to the concepts. We formulate such a comparative procedure into a novel information-regularized textual inversion. We observe that, due to the ambiguity of the comparative formulation, optimizing a continuous concept vector does not converge to desirable solutions. To find a human interpretable and meaningful concept, we parameterize the concept by discrete embeddings with existing language tokens as prior. This reduces the risk of discovered concepts being optimized to adversarial concepts [3] and makes them reside in a meaningful embedding space. Additionally, since the discrete parameterization restricts the expressivity, we propose the second refinement stage, where we learn the residual continuous concept vectors to better fit the discovered concepts by leveraging the discovered ones. To better shape the residual concept, we generate the synthetic images from each concept and optimize the residual concepts in the continuous embedding space. It helps the discovered concepts further align with given objects. This `DISCOD` (DIstinct and Shared COncept Discovery) procedure is illustrated in Fig. 1.

We validate our method of discovering commonalities and differences given respective image sets of two objects and three tasks. In our systematic experiment, we demonstrate that `DISCOD` effectively discovers the shared and distinct concepts. We show that `DISCOD` outperforms baselines in discovering these concepts with respect to CLIPScore and discovery success rate. We also compute the alignment and agreement scores by human study, where the subjects agree that the discovered concepts represent the percieved concepts well. Additionally, we demonstrate the applicability of `DISCOD` in three applications: image editing, fine-tuning for personalization, and group bias mitigation. Our method can be integrated with the existing text-conditioned image editing. During fine-tuning diffusion models given a few images, `DISCOD` reduces the undesirable entanglement. In a recognition scenario, where the classes are correlated to their attributes, inducing short-cut learning, our method can find the group bias and enables us to mitigate bias. The effectiveness of our bias mitigation is validated on Waterbirds (Sagawa et al., 2020) and CelebA (Liu et al., 2015).

## 2 METHOD

In this section, we first provide a background of the prior methods that discover concepts representing a single object. Then, we introduce our goal to simultaneously discover shared and distinct concepts given two objects and propose our method, `DISCOD`, to discover these concepts effectively.

**Background: Inversion-based concept discovery.** Prior concept discovery methods with vision language models (VLMs) (Rombach et al., 2022; Radford et al., 2021) aim to identify or optimize a concept given a few images containing a single object by applying model inversion techniques to discover its textual concept embedding; we call it as inversion-based concept discovery. Textual inversion (TI) (Gal et al., 2023) is a seminal work in this context; they optimize a few newly initialized

---

[3]The concepts are over-fitted in arbitrary ways, rather than containing appropriate concepts.

tokens that correspond to the new concept while the original pre-trained tokens remain the same. Once a concept representing the object is discovered, its textual token can be used for image synthesis and style transfer. Some works (Safaee et al., 2024; Ruiz et al., 2023; Sohn et al., 2023; Avrahami et al., 2023; Huang et al., 2023b; Kumari et al., 2023) have further developed techniques for image synthesis, including personalization and text-guided synthesis. Another important research direction is to decompose a given object into sub-concepts (Vinker et al., 2023; Chefer et al., 2024). It is useful to understand how VLMs recognize the object and can improve our lack of understanding of VLMs. This direction provides a way to gain insights into VLMs' representation mechanisms.

While it is useful to extract the shared concept given an object, it is even more crucial to discover both the shared and distinct concepts; this discovery can further provide a richer understanding of how VLMs recognize concepts between objects. Since the previous methods optimize for a single object, their methods cannot be directly applied or extended to discover the shared and distinct concepts. Their discovered textual token can be adversarial; certain textual tokens can encode a large coverage of information, making other textual token concepts meaningless. In this work, we tackle this problem by introducing information-regularized textual inversion.

## 2.1 DISCOD: DIstinct and Shared COncept Discovery

We focus on discovering the shared and distinct concepts given two objects. We first introduce some notations. Let $A$ and $B$ be two image sets, where each set contains distinct concepts, $\mathbf{y}_{A \setminus B}$ and $\mathbf{y}_{B \setminus A}$, compared to another set and also shares a common concept, $\mathbf{y}_{A \cap B}$, between sets. Our goal is to discover $\mathbf{y}_{A \setminus B}, \mathbf{y}_{B \setminus A}$, and $\mathbf{y}_{A \cap B}$ such that they sufficiently represent the given objects as follows:

$$\min_{\substack{\mathbf{y}_{A \setminus B}, \mathbf{y}_{B \setminus A}, \\ \mathbf{y}_{A \cap B}}} \mathcal{L}_s, \text{ where } \mathcal{L}_s = - \left[ I\left(A \mid \mathbf{y}_{A \setminus B}, \mathbf{y}_{A \cap B}\right) + I\left(B \mid \mathbf{y}_{B \setminus A}, \mathbf{y}_{A \cap B}\right) \right], \quad (1)$$

where $I\left(\cdot\right)$ is mutual information. However, we find that the above objective is insufficient to separate $\mathbf{y}_{A \setminus B}$ and $\mathbf{y}_{B \setminus A}$ from $\mathbf{y}_{A \cap B}$ since a concept often leaks to another concept (*i.e.*, failure to optimize $\mathbf{y}_{A \setminus B}$ or $\mathbf{y}_{B \setminus A}$), or $\mathbf{y}_{A \cap B}$ can easily be optimized to representative non-relative concepts (*i.e.*, failure to separate $\mathbf{y}_{A \cap B}$; trivial solutions exists). For example, let $A$ be `yellow chair` and $B$ be `yellow table`. Then, we desire $\mathbf{y}_{A \setminus B} = $ `chair`, $\mathbf{y}_{B \setminus A} = $ `table`, and $\mathbf{y}_{A \cap B} = $ `yellow`. However, there is still a potential solution that $\mathbf{y}_{A \setminus B} = $ `yellow chair`, $\mathbf{y}_{B \setminus A} = $ `yellow table`, and $\mathbf{y}_{A \cap B} = $ `photo`. To prevent this suboptimal solution, we introduce the information bottleneck (Tishby & Zaslavsky, 2015; Gilad-Bachrach et al., 2003) that restricts the representation space of each concept as follows:

$$\min_{\substack{\mathbf{y}_{A \setminus B}, \mathbf{y}_{B \setminus A}, \\ \mathbf{y}_{A \cap B}}} \mathcal{L}_s + \lambda_m \mathcal{L}_m, \text{ where } \mathcal{L}_m = \left[ I(A \mid \mathbf{y}_{A \setminus B}) + I(B \mid \mathbf{y}_{B \setminus A}) + I(A, B \mid \mathbf{y}_{A \cap B}) \right]. \quad (2)$$

The second term, $\mathcal{L}_m$, reduces the representation complexity and prevents concepts from containing unnecessary information. This reduces the overlapping semantics between them; thereby, the shared concept between them could be maximized. However, this objective function is difficult to directly optimize; thus, we relax the problem. In the following section, we introduce the two-stage framework: (1) we propose relaxed information regularization methods in the first stage, and (2) we refine the concepts to represent two image sets in the second stage.

## 2.2 First stage: Information-regularized textual inversion

In the first stage, we aim to discover the distinct concepts of $\mathbf{y}_{A \setminus B}$ and $\mathbf{y}_{B \setminus A}$ while maximizing the separation of shared concepts between them to be incorporated into $\mathbf{y}_{A \cap B}$ by Eq. (2). Due to the difficulty of directly solving Eq. (2), we propose the following relaxation techniques.

**Textual token embedding.** We parameterize $\mathbf{y}_{A \setminus B}, \mathbf{y}_{B \setminus A}, \mathbf{y}_{A \cap B}$ in the discrete embedding space $\mathcal{E}$, which is the pre-trained text token embedding, rather than in the continuous space. This parameterization provides an upper bound of Eq. (1), because we have $\inf_{\mathbf{y}} -I\left(\mathbf{x} \mid \mathbf{y}\right) \leq \inf_{\mathbf{y} \in \mathcal{E}} -I\left(\mathbf{x} \mid \mathbf{y}\right),$[4] where $\mathbf{x}$ and $\mathbf{y}$ be images and texts. This optimization is akin to an upper bound minimization (Hunter & Lange, 2004) simplifying the optimization as long as the upper bound is easier to optimize in practice. Also, since the pre-trained text tokens embed vast prior knowledge of human interpretable language, this implicitly acts as a prior.

---

[4]Derived from $\sup_{\mathbf{y} \in \mathcal{E}} I\left(\mathbf{x} \mid \mathbf{y}\right) \leq \sup_{\mathbf{y}} I\left(\mathbf{x} \mid \mathbf{y}\right)$

**Relaxed objective function.** The maximization of the mutual information can be expressed by the conditional entropy, leading us to maximize the conditional probability. Simply, our approximation results in the cosine similarity between the image sets and the text embedding (refer to the derivation in the appendix). In the first stage, we relax $\mathcal{L}_s$ with the negative cosine similarity, denoted as $\hat{\mathcal{L}}_s$. To compute the cosine similarity, we use CLIP (Radford et al., 2021) for representing image and text embeddings. For mapping the concept vectors to text embeddings, with the discrete parameterization, we concatenate $\mathbf{y}_{A\setminus B}, \mathbf{y}_{B\setminus A}$ with $\mathbf{y}_{A\cap B}$ as $[\mathbf{y}_{A\cap B}; \mathbf{y}_{A\setminus B}]$ and $[\mathbf{y}_{A\cap B}; \mathbf{y}_{B\setminus A}]$. Then, we extract these respective text embeddings using the CLIP text encoder. The relaxation $\hat{\mathcal{L}}_s$ is as follows:

$$\hat{\mathcal{L}}_s(A, B, \mathbf{y}_{A\setminus B}, \mathbf{y}_{B\setminus A}, \mathbf{y}_{A\cap B}) = \left(1 - \texttt{CosSim}\left(\texttt{CLIP}_I(A), \texttt{CLIP}_T\left([\mathbf{y}_{A\cap B}; \mathbf{y}_{A\setminus B}]\right)\right)\right) +$$
$$\left(1 - \texttt{CosSim}\left(\texttt{CLIP}_I(B), \texttt{CLIP}_T\left([\mathbf{y}_{A\cap B}; \mathbf{y}_{B\setminus A}]\right)\right)\right), \quad (3)$$

where $\texttt{CosSim}(\cdot)$ denotes the cosine similarity, and $\texttt{CLIP}_{\{I,T\}}$ denotes CLIP image/text encoders.

The information regularization applied to the distinct concepts is also approximated by the cosine similarity. This time, we compute the cosine similarity between the discrete embedding space $\mathcal{E}$ and $\{\mathbf{y}_{A\setminus B}, \mathbf{y}_{B\setminus A}\}$ (refer to the derivation in the appendix), *i.e.*, $\texttt{CosSim}\left(\mathcal{E}, \mathbf{y}_{A\setminus B} \text{ or } \mathbf{y}_{B\setminus A}\right)$. We apply the softmax operation to get the probabilistic distribution over the cosine values. We can compute the KL divergence between this distribution and the uniform distribution $U$ by the cross-entropy. Our regularization is as follows:

$$\hat{\mathcal{L}}_m(\mathcal{E}, \mathbf{y}_{A\setminus B}, \mathbf{y}_{B\setminus A}) = \texttt{CE}\left(p\left(\texttt{CosSim}\left(\mathcal{E}, \mathbf{y}_{A\setminus B}\right)\right), U\right) + \texttt{CE}\left(p\left(\texttt{CosSim}\left(\mathcal{E}, \mathbf{y}_{B\setminus A}\right)\right), U\right), \quad (4)$$

where $P(\cdot)$ denotes the softmax function, and $U$ the uniform distribution. We adopt PEZ (Wen et al., 2024) to optimize $\hat{\mathcal{L}}_s + \lambda_m \hat{\mathcal{L}}_m$ on the discrete embedding.

### 2.3 SECOND STAGE: FINE-TUNING WITH SYNTHETIC CONCEPTS

Although we find meaningful concepts from the first stage, it is not sufficient to align the concepts with the given two objects, $A$ and $B$, due to the limited expressivity of the discrete representation. To refine, we learn their continuous residual concepts in this stage. We generate synthetic images with the learned discrete tokens in the first stage using text-to-image (T2I) diffusion models. We denote the synthetic images as $\overline{A \setminus B}, \overline{B \setminus A}$, and $\overline{B \cap A}$, respectively. We use these synthetic images to optimize $\mathbf{y}_{A\setminus B}, \mathbf{y}_{B\setminus A}$, and $\mathbf{y}_{A\cap B}$ in the continuous embedding. Synthetic images are helpful for preventing converging non-meaningful concepts during fine-tuning. We also use the discovered concepts from the first stage as initialization for the second stage. To optimize further, we use the T2I diffusion models; $p(\mathbf{x}|\mathbf{y}) \sim \mathcal{N}(\alpha_t \mathbf{x}, \sigma_t^2)$ where coefficient $\alpha_t$ and $\sigma_t$ satisfy $p(\mathbf{x}|\mathbf{y}) \sim \mathcal{N}(\mathbf{0}, \mathbf{1})$ at $t = 0$. Thus, in the second stage, we relax the maximization of $I(\mathbf{x} \mid \mathbf{y})$ with the minimization of the diffusion loss as $\mathcal{L}_d(A, \mathbf{y}) = ||\boldsymbol{\epsilon} - \boldsymbol{\epsilon}_\theta(\mathbf{x}, t, \mathbf{y})||_2^2$, where $\mathbf{x} \in A$. The final optimization is as follows:

$$\min_{\substack{\mathbf{y}_{A\setminus B}, \mathbf{y}_{B\setminus A}, \\ \mathbf{y}_{A\cap B}}} \hat{\mathcal{L}}_s, \quad \text{where } \hat{\mathcal{L}}_s = \mathcal{L}_d(A, [\mathbf{y}_{A\cap B}; \mathbf{y}_{A\setminus B}]) + \mathcal{L}_d(B, [\mathbf{y}_{A\cap B}; \mathbf{y}_{B\setminus A}])$$
$$+ \mathcal{L}_d(\overline{A \setminus B}, \mathbf{y}_{A\setminus B}) + \mathcal{L}_d(\overline{B \setminus A}, \mathbf{y}_{B\setminus A}) + \mathcal{L}_d(\overline{B \cap A}, \mathbf{y}_{A\cap B}). \quad (5)$$

By fine-tuning, we discover the concepts aligned with the given two image sets. We discard the regularization term $\hat{\mathcal{L}}_m$, and instead we use synthetic images, which provide implicit regularization. The derivation of the objective can be found in the appendix. Figure 1 shows our two-stage framework.

## 3 EXPERIMENTS

In this section, we conduct experiments on various tasks: discovering both commonalities and differences through textual inversion (Sec. 3.1), applying `DISCOD` for editing task (Sec. 3.2), fine-tuning Text-to-Image (T2I) Diffusion models for personalization (Sec. 3.3), and mitigating group bias in the Waterbirds and CelebA datasets (Sec. 3.4).

### 3.1 COMMONALITY & DIFFERENCE TEXTUAL INVERSION

**Qualitative results.** We curate the real image pairs from Unsplash [5]. Additionally, we utilize the DreamBooth dataset (Ruiz et al., 2023). We use Stable Diffusion 2.1-base for experiments to discover both commonalities and differences by textual inversion given the two image sets.

---
[5]https://unsplash.com/

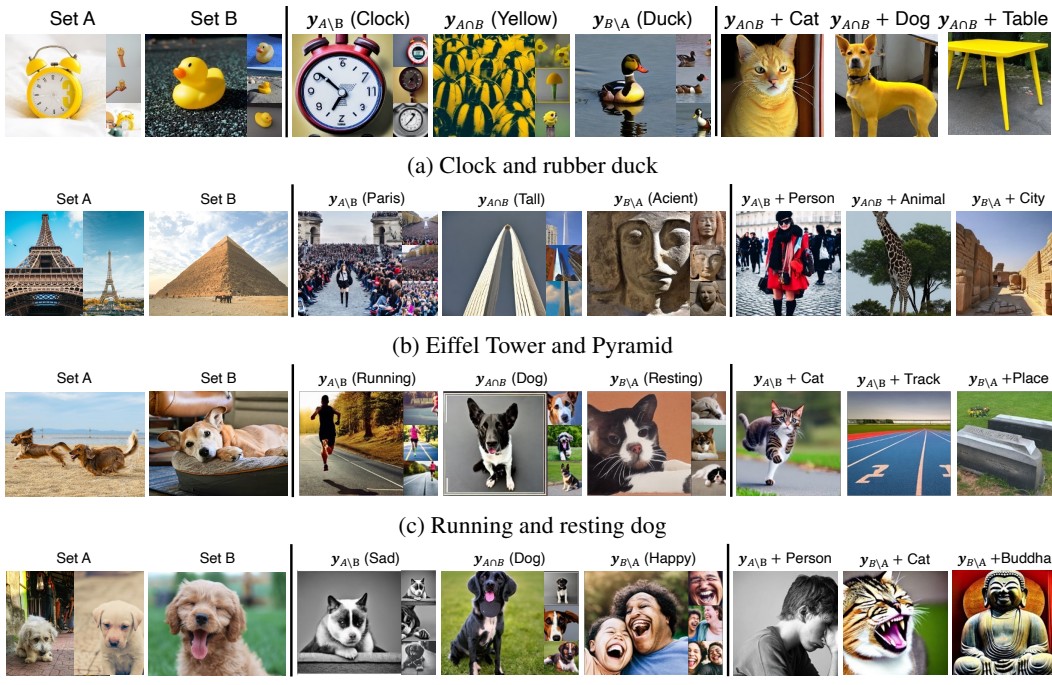

(a) Clock and rubber duck

(b) Eiffel Tower and Pyramid

(c) Running and resting dog

(d) Sad and happy dog

Figure 2: **Qualitative on real pairs.** We apply DISCOD to the real pairs from Set 1 and Set 2. We generate images from the discovered concepts denoted as $\mathbf{y}_{A \setminus B}, \mathbf{y}_{B \setminus A}, \mathbf{y}_{A \cap B}$ and the discovered concepts with additional text tokens, $e.g.$, $\mathbf{y}_{A \cap B}$ + Cat where + is the concatenation operation. We show the corresponding words next to the discovered concepts for convenience of referencing.

Figure 2 shows the qualitative results. We apply DISCOD to real pairs to validate that DISCOD discovers meaningful commonalities and differences. Specifically, after optimizing the common and distinct concepts between two objects, we generate images from the T2I diffusion model with the prompts, "the photo of $\mathbf{y}_{A \setminus B}, \mathbf{y}_{A \cap B}$ or $\mathbf{y}_{B \setminus A}$", and with the additional text tokens, $e.g.$, "a photo of a $\mathbf{y}_{A \cap B}$ cat." It helps us determine whether the discovered concepts are valid.

As shown in Fig. 2a, we give the clock and rubber duck, both of which share a yellow color. We can interpret $\mathbf{y}_{A \cap B}$ as yellow, and $\mathbf{y}_{A \setminus B}$ and $\mathbf{y}_{B \setminus A}$ represent clock and duck, respectively. Our method successfully discovers $\mathbf{y}_{A \cap B}$ as yellow because $\mathbf{y}_{A \cap B}$ generates the shared color as shown in 2a. The discovered $\mathbf{y}_{A \setminus B}$ and $\mathbf{y}_{B \setminus A}$ align with their respective categories, as expected. In the case of the Eiffel Tower and the Pyramid (See Fig. 2b), the discovered $\mathbf{y}_{A \setminus B}, \mathbf{y}_{A \cap B}, \mathbf{y}_{B \setminus A}$ represents Paris, tall, and ancient, respectively. Since both the Eiffel Tower and the Pyramid are tall structures, the discovered concept $\mathbf{y}_{A \cap B}$ is reasonable.

We also provide examples involving two different dogs in distinct poses (See Fig. 2c). The commonality $\mathbf{y}_{A \cap B}$ is identified as dog. $\mathbf{y}_{A \setminus B}$ is related to running as the individual and composite images depict a running person and a running track. $\mathbf{y}_{B \setminus A}$ represents resting, as indicated by the resting pose and the resting place in the images. Finally, we apply DISCOD to a pair describing different emotional states: one negative and the other positive. The discovered $\mathbf{y}_{A \setminus B}$ and $\mathbf{y}_{B \setminus A}$ capture these emotional states. $\mathbf{y}_{A \setminus B}$, when combined with a person, shows a negative situation, while $\mathbf{y}_{B \setminus A}$, when combined with Buddha, shows a smiling Buddha.

**Attention Map.** We visualize the cross-attention map between the discovered concepts and images by running DDIM inversion and computing the cross-attention scores.

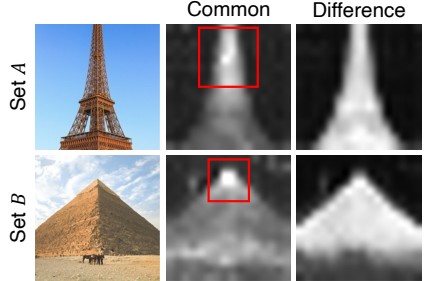

Figure 3: **Attention map visualization.** We visualize the cross-attention map between the discovered concepts and the training images.

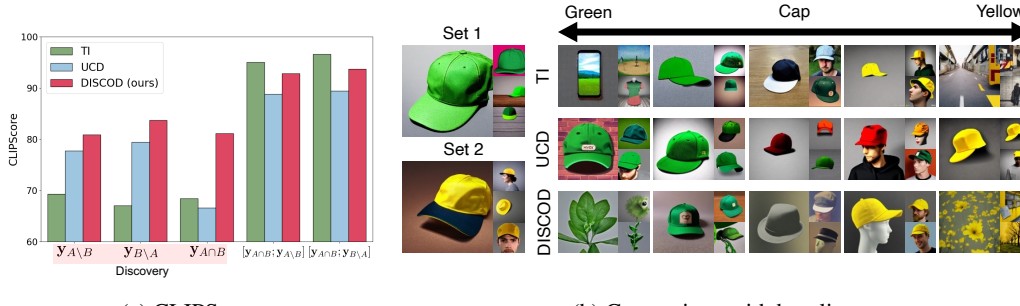

(a) CLIPScore             (b) Comparison with baselines

Figure 4: **CLIPScore and qualitative result on the synthetic data.** (a) We measure the CLIPScore between the known concepts used in synthetic data generation and the images generated from the discovered concepts. (b) We generate images based on the concepts discovered through our method. The arrow above the images indicates the desired concept. The middle of the two concepts represents the compositional concept of two concepts.

The salient regions of the attention map are highlighted with red boxes as shown in Fig. 3. The common concept "Tall" focuses on the upper portions of the provided images, indicating that the discovered concepts capture meaningful factors. The result shows that the discovered concepts correspond to relevant areas within the images.

**Quantitative result - CLIPScore.** To enable quantitative comparisons, we generate synthetic data in a controlled environment by using T2I diffusion models. We generate image pairs with the prompts "the photo of $\mathbf{y}_{A \cap B}$ {$\mathbf{y}_{A \setminus B}$ or $\mathbf{y}_{B \setminus A}$}," where a pair contains only one shared concept. We choose Textual Inversion (TI) (Gal et al., 2023; Vinker et al., 2023) and Unsupervised Concept Discovery (UCD) (Liu et al., 2023a) as baselines and modify these methods to discover commonalities and differences between two objects. Vinker et al. (2023) have introduced a method for decomposing an individual instance into sub-concepts using a binary tree structure of text tokens. Liu et al. (2023a) have proposed an unsupervised approach for discovering concepts from image collections. To ensure that commonalities are captured across each image set, we set the weight combination coefficient to 0.5 for each concept token.[6]

Given the controlled nature of the synthetic pairs, we know the commonalities and differences between them. We compute the CLIPScore (Hessel et al., 2021) between the known concept (text) and the images generated from the discovered concepts. Following the approach of Hessel et al. (2021), we scale the scores by a factor of 2.5. If a discovered concept effectively represents its respective concept, the CLIPScore is high. Figure 4a shows the CLIPScore of each discovered concept and compositional concept. Although TI exhibits a high similarity in $[\mathbf{y}_{A \cap B}; \mathbf{y}_{A \setminus B}]$ and $[\mathbf{y}_{A \cap B}; \mathbf{y}_{B \setminus A}]$, our method (DISCOD) achieves a higher score than the baselines for $\mathbf{y}_{A \setminus B}$, $\mathbf{y}_{B \setminus A}$, and $\mathbf{y}_{A \cap B}$. Figure 4b have a similar tendency with the CLIPScore results. Thus, DISCOD shows more fine-grained concept discovery capabilities than baselines.

**Quantitative result - Success rate.** We also directly measure the success rate of discovery. We conduct a human study where each participant answers 60 questions. After observing a high Pearson correlation of 0.87 between participants and experts, we assess the success and failure rates based on expert evaluations. As shown in Table 1, DISCOD surpasses other baselines in terms of the success rate for decomposition. Since TI shows high compositional performance, the limitations in discovery are not due to fitting issues. This validates the effectiveness of our approach in uncovering concepts within images.

Table 1: **Success rate on synthetic data.** Discovery means $\mathbf{y}_{A \setminus B}, \mathbf{y}_{B \setminus A}, \mathbf{y}_{A \cap B}$, and Comp. means $\mathbf{y}_A, \mathbf{y}_B$.

| Model | Success rate | | |
|---|---|---|---|
| | Discovery | Comp. | Mean |
| TI | 0.42 | **0.72** | 0.58 |
| UCD | 0.36 | 0.60 | 0.48 |
| DISCOD | **0.77** | 0.60 | **0.69** |

**Ablation Study - KL divergence.** We perform an ablation study on our proposed regularization, which is an information bottleneck in the first stage. Figure 5a show the CLIPScore and the

---

[6]Further implementation details are provided in the appendix.

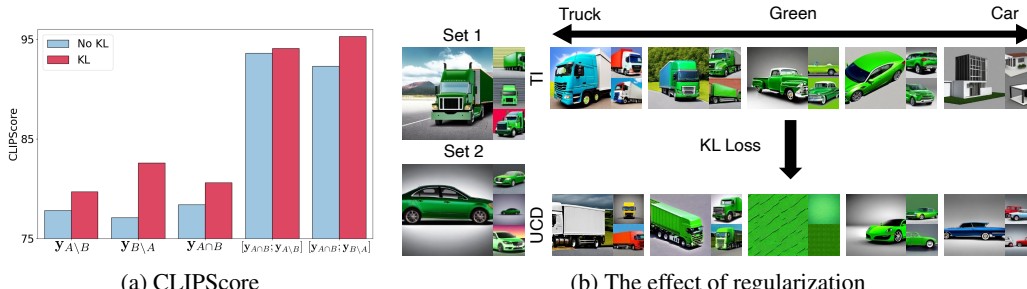

(a) CLIPScore  (b) The effect of regularization

Figure 5: **CLIPScore and qualitative result with and without regularization.** We measure the CLIPScore and generate the images in the first stage, both with and without regularization. Our observations indicate that regularization enhances discovery performance and makes discovered concepts optimized toward meaningfulness.

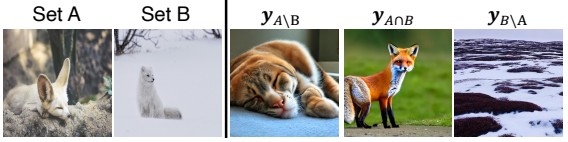

Figure 7: **Example used for alignment and aggrement.** We apply DISCOD to the pair of fennec and arctic fox. DISCOD discover resting pose, fox, and arctic for $\mathbf{y}_{A \setminus B}, \mathbf{y}_{A \cap B}$, and $\mathbf{y}_{B \setminus A}$.

Table 2: **Alignment and agreement of fennec and arctic fox.** Alignment of $\mathbf{y}_{A \setminus B}$ is low because participants thought it as the desert. However, participants agree the pose is $\mathbf{y}_{A \setminus B}$.

| Concept | Alignment | Aggrement |
|---|---|---|
| $\mathbf{y}_{A \setminus B}$ | 0.28 | 0.75 |
| $\mathbf{y}_{A \cap B}$ | 0.95 | 0.95 |
| $\mathbf{y}_{B \setminus A}$ | 0.86 | 0.97 |

generated images of DISCOD with and without regularization. It shows that regularization improves the CLIPScore, indicating the regularization's necessity for making the concepts more meaningful. Figure 5b shows the regularization effect. In this example, the common concept is green, while the distinct concepts are truck and car. Without the KL divergence loss, the shared concept, $\mathbf{y}_{A \cap B}$, generates a green car (See the above row in Fig. 5b). Since "green car + truck" could potentially describe green Truck, it is understandable but is not desirable. Furthermore, the distinct concept of $\mathbf{y}_{B \setminus A}$ lacks meaningfulness. After applying our regularization, the discovered concepts become noticeably more meaningful (See the bottom row of Fig. 5b). Thus, regularization is effective.

**Ablation study - Second stage.** The second stage is designed to better align the representation with the given objects. Figure 6 shows the first stage generates a structure resembling a pyramid; however, it also encompasses various other concepts due to the restriction to discrete token embeddings. In the second stage, the concepts align more accurately with the pyramid than that of the first stage, validating the effectiveness of the second stage.

**Human study - Alignment and agreement.** The discovered concepts can be different from what we perceptually expect. We apply DISCOD to the pair of fennec (desert) and arctic fox. We expect the shared concept is a fox, and the distinct concepts are desert and arctic, reflecting their differing habitats. However, $\mathbf{y}_{A \setminus B}$ of fennec fox is more related to pose rather than the desert, specifically characterized by the crouching posture. This is primarily because the majority of fennec foxes are observed in a crouching position.

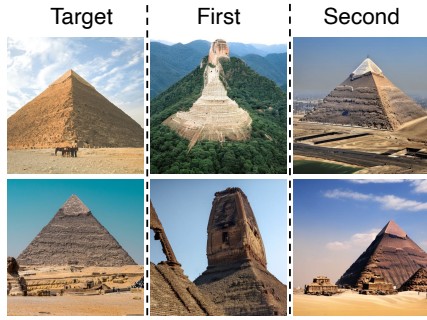

Figure 6: **Ablation study of the second stage.** The second stage optimizes the discovered concept further to represent the given object.

We conduct a human study to explore alignment and agreement. Participants are asked to write short answers describing the commonalities and differences between a pair of images. Afterward, we show them the predicted concepts from DISCOD, and they compare their written short answers with the predicted concepts from DISCOD. For alignment, we quantify the response by assigning a score of 1 if the participant answer that the predicted concept aligned with their own. The generated

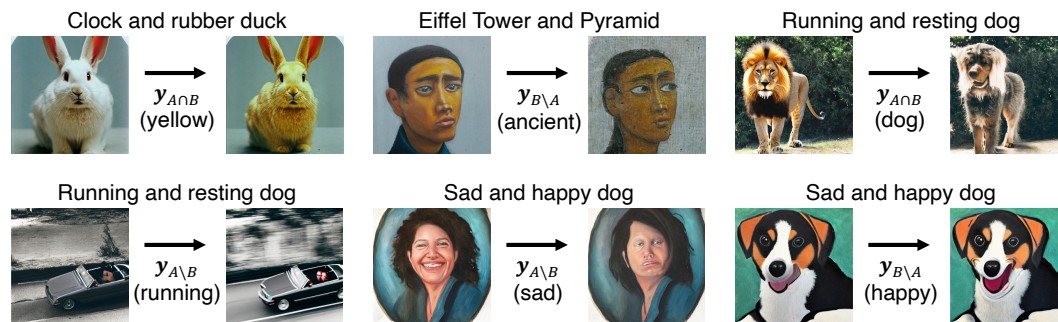

Figure 8: **Editing results with the discovered concepts** We apply the editing method, Prompt-to-Prompt (Hertz et al., 2022), to the given image. It validates the meaning of the discovered concepts in Fig. 2. It also provide the possible application of ours.

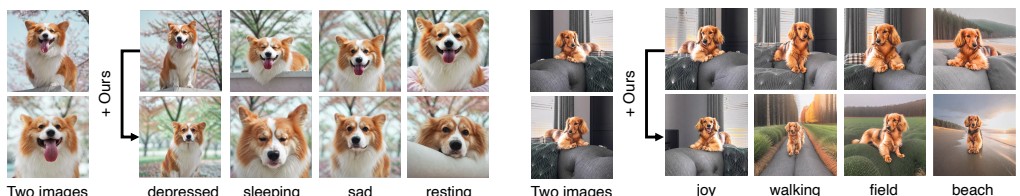

Figure 9: **Bias mitigation in DreamBooth.** We apply our method to DreamBooth (Ruiz et al., 2023). Given two images, we discover the concepts between them. The top row shows DreamBooth without DISCOD, while the bottom row show DreamBooth with DISCOD. From the left example, we discover a common concept of emotion, and from the right samples, we discover the common concept of location. Ours improves the result of DreamBooth.

image may still represent concepts that they initially overlooked, although participants' answers differ. Participants are also asked whether the generated image from the discovered concepts represents the commonalities or differences. Their responses are rated on a scale from 1 to 5, with higher scores indicating that the discovered concept accurately represents the relevant concept. We normalize the agreement value. The key difference from alignment is that agreement evaluates whether the discovered concepts from DISCOD are understandable.

The alignment score for the fennec fox's attributes is low, as participants predominantly mention "desert", as expected. Some responses, however, relate to the fox's pose and resting behavior. Consequently, the concepts associated with pose are easily recognizable, resulting in a high agreement score. Other human studies can be found in the appendix.

## 3.2 IMAGE EDITING

Image editing based on text conditions offers high user control without requiring specific image editing skills. Users simply provide an image and a descriptive text condition. In this section, we combine Prompt-to-Prompt (Hertz et al., 2022) with DISCOD. Using the concepts discovered in Fig. 2, we apply them to the Prompt-to-Prompt framework, allowing us to evaluate the validity and effectiveness of the discovered concepts.

As shown in Fig. 8, we observe that the concept $\mathbf{y}_{A \cap B}$ of a clock and a rubber duck transforms a rabbit into a yellow rabbit. The concept $\mathbf{y}_{B \setminus A}$, which merges the Eiffel Tower and a pyramid, transforms the painting to an aged appearance. Additionally, $\mathbf{y}_{A \setminus B}$ of the running concept gives a blur effect to the car, conveying the idea of motion. Other examples involving objects and emotions also align well with the identified concepts. These results validate the effectiveness of our method and its potential application in image editing.

## 3.3 DISENTANGLEMENT IN TEXT-TO-IMAGE PERSONALIZATION

Text-to-Image (T2I) personalization aims to generate personalized images, given a few images of the target instance. Ruiz et al. (2023) proposed DreamBooth, a method that fine-tunes T2I diffusion

Table 3: **Prompt performance on Waterbirds and CelebA.** We report worst-group and average accuracy with their gap. Zero-shot Prompt only uses class labels, Group Prompt exploits the knowledge of biased information, and B2T uses the discovered keyword. DISCOD uses the discovered distinct concepts. **Bold** and Underline are the best and the second best, respectively.

| Dataset | Method | RN50 | | | ViT-B-32 | | | ViT-H-14 | | |
|---|---|---|---|---|---|---|---|---|---|---|
| | | Worst | Avg | Gap | Worst | Avg | Gap | Worst | Avg | Gap |
| Waterbirds | Zero-shot Prompt | 44.2 | 70.2 | 26.0 | 47.3 | 71.5 | 24.2 | 37.2 | 84.0 | 46.8 |
| | Group Prompt | 52.6 | **79.3** | 26.7 | **60.1** | **79.4** | 19.3 | 34.6 | 84.8 | 50.2 |
| | B2T | 57.2 | 75.0 | **17.8** | 57.8 | 75.9 | 18.1 | 35.5 | 84.7 | 49.2 |
| | DISCOD (ours) | **59.3** | 77.5 | 18.2 | 58.3 | 76.2 | **17.9** | **39.9** | **85.3** | **45.4** |
| CelebA | Zero-shot Prompt | 74.0 | 83.7 | 9.7 | 78.9 | 90.4 | 11.5 | 45.2 | 88.3 | 43.1 |
| | Group Prompt | 78.3 | **87.8** | 9.5 | 82.8 | 90.4 | 7.6 | 46.0 | 88.9 | 42.9 |
| | B2T | **79.0** | 86.3 | 7.3 | 85.0 | 89.2 | 4.2 | 48.8 | 89.0 | 40.2 |
| | DISCOD (ours) | 78.1 | 85.5 | 7.4 | **86.7** | 88.6 | **1.9** | **60.2** | **89.4** | **29.2** |

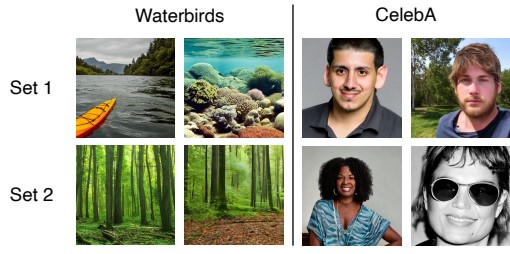

Figure 10: **Visualization of discovered bias.** We visualize the discovered biases, $\mathbf{y}_{A \setminus B}$ and $\mathbf{y}_{B \setminus A}$, from Waterbirds and CelebA. The discovered biases represent their known biases well.

Table 4: **Ablation study of CelebA.** Male is a well-known bias in CelebA, thus more effective in Group Prompt than others. Our discovered concept corresponding to male images is also effective, which is the same for Group Prompt. **Bold** is the best.

| Method | Worst | Avg | Gap |
|---|---|---|---|
| Group Prompt (Male) | **78.3** | 87.7 | 9.5 |
| Group Prompt (Female) | 76.7 | 88.4 | 11.7 |
| Group Prompt (Both) | 76.9 | 89.6 | 12.2 |
| DISCOD (Male) | **82.2** | 86.4 | 4.5 |
| DISCOD (Female) | 70.2 | 82.6 | 12.4 |
| DISCOD (Both) | 78.1 | 85.5 | 7.3 |

models using a few images for personalization. However, this approach is vulnerable to entanglement if given images are biased to some attributes, such as facial expression or location. For example, if most reference images have a smiling face, the generated images may consistently show a smiling expression, even when the prompt implies a negative emotion. To address this issue, we propose to combine DISCOD with DreamBooth.

We first discover the shared and distinctive concepts by applying DISCOD in the first stage. We fine-tune the diffusion model by providing the specific prompt, *e.g.*, "the photo of $\mathbf{y}_{A \cap B}$ $\mathbf{y}_{A \setminus B}$ object", rather than "the photo of object". This detailed description helps disentangle the undesirable concepts as shown in SDI (Kim et al., 2024a). For this experiment, we use Stable Diffusion XL and LoRA (Hu et al., 2022).

Figure 9 shows the results of applying DreamBooth with and without our method. In the left example of Fig. 9, the model exhibits a bias toward emotions such as happiness or joy, which leads to the personalized model generating the dog instances with its tongue out. See the bias result of DremBooth given depressed, sleeping, sad, and resting prompts. In contrast, our method enables the model to generate diverse emotional states, so that the dog instances are generated without its tongue out. The right example in Fig. 9 has the location bias. While the naive DreamBooth generates images with limited backgrounds, our approach generates images with varied backgrounds. Thus, our discovery methodology is effective in tackling the entanglement during fine-tuning.

## 3.4 MITIGATING GROUP BIAS

Does recognizing the commonality and differences between sets help VLMs recognize objects in reverse? The neural network is vulnerable to bias. For example, the waterbird class in the Waterbirds dataset is often in the water. The waterbird with the land background shows lower accuracy compared

to the average accuracy. We apply `DISCOD` to discover the concepts between the mispredicted set of two classes like B2T (Kim et al., 2024b). We denote "`the photo of {class}`" as a Zero-shot Prompt (Radford et al., 2021) and "`the photo of {class} in the {group}`" as Group Prompt (Zhang & Re, 2022). B2T uses the group labels as their discovered keywords from midpredicted images. We use the pre-trained ResNet50 of CLIP for `DISCOD` and use the discovered distinct concepts in the first stage as group labels.

We evaluate the methods on Waterbirds and CelebA datasets. We report the worst-group and average accuracy with their gap (See Table 3). Compared to Zero-shot Prompt, `DISCOD` improves the worst-group accuracy. In addition, our discovered concepts are effective compared to Group Prompt and B2T. Especially in ViT-H-14, ours achieves the highest score in worst-group, average, and their gap.

The bias of Waterbirds and CelebA are the background and gender, respectively. We visualize the discovered concepts (See Fig. 10). Our discovered distinct concepts are related to water and forest in the Waterbirds dataset; the ones are male and female in CelebA. The visualization shows that `DISCOD` discovers the meaningful bias. We experiment with the ablation to incorporate one or both of our discovered concepts into the prompting. We use male bias for Group Prompt because the known group is effective bias (See the performance of Group Prompt in Table 4). Among the discovered concepts, the one generating the male image is more effective rather than the female and both, which is the same tendency to Group Prompt.

## 4 RELATED WORK

**Inversion-based concept discovery.** With VLMs (Radford et al., 2021; Desai et al., 2023; Girdhar et al., 2023; Jia et al., 2021; Li et al., 2022; Rombach et al., 2022; Ramesh et al., 2021; Kang et al., 2023; Chen et al., 2024a), the concept discovery aims to identify the concept corresponding with a few images containing a single object. Textual inversion (TI) (Gal et al., 2023) optimizes the textual embedding to the given object using T2I diffusion models. Subsequently, recent works (Vinker et al., 2023; Chefer et al., 2024) have proposed methods for decomposing a single object into sub-concepts. Vinker et al. (2023) have introduced the tree-structure construction of token embedding, uncovering the hidden sub-concepts of the single object. Chefer et al. (2024) also have decomposed a single concept into sub-concepts to understand the internal representation of VLMs. In this work, we focus on discovering both the commonalities and differences between two image sets, in contrast to prior research, which focuses on discovering commonalities in an image set. We develop information-regularized methods tailored to our objective.

**Explainable machine learning.** Explainable machine learning aims to provide terms understandable to humans about machine decision (Doshi-Velez & Kim, 2017). Prior works have proposed algorithms to measure the score that affects the decision of models and provide saliency map (Kim et al., 2021; Choe et al., 2022; Lee et al., 2022; Selvaraju et al., 2017; Li et al., 2016; Arras et al., 2017). Additionally, neural networks accumulate their knowledge into neurons referred to as knowledge neuron (Dai et al., 2022). Some works (Liu et al., 2023b; Dai et al., 2022; Chen et al., 2024b) have explored which neurons influence model decisions. Concept-based models (Koh et al., 2020; Yuksekgonul et al., 2023; Zhou et al., 2018) are composed of the concept bottleneck layer, where each layer represents human interpretable concepts. In this work, `DISCOD` identifies the shared and distinct concepts recognized by VLMs and visualizes the concepts in the human interpretable medium.

## 5 CONCLUSION

We take a closer look at how VLMs recognize the commonalities and differences between two objects. Unlike the previous works that focus on single objects, `DISCOD` discovers shared and distinct concepts simultaneously. We formulate the task to maximize the information of sub-concepts for the given objects and propose a two-stage framework. We validate `DISCOD` on pairs of real and synthetic settings, observing that `DISCOD` identifies various concepts like color, category, and abstract concepts. Both CLIPScore evaluations and human studies demonstrate the effectiveness of `DISCOD`. In addition, we also validate the effectiveness of `DISCOD` on three tasks: image editing, fine-tuning diffusion models on the DreamBooth dataset, and group-bias mitigation on WaterBirds and CelebA. We believe that understanding how machines distinguish between objects will lead to a better understanding of machines, which in turn leads to better performance.

## ETHICS STATEMENT

`DISCOD` discovers the shared and distinct concepts between two sets of images and provides them with a form that humans can understand. Thus, we can use it to investigate the two objects to help our understanding of these objects. However, if we rely heavily on the discovered ones, it leads humans to misunderstand these objects because the discovered ones are related to how VLMs perceive them; it can have a negative societal impact. Thus, we should use it as a tool to help us. In terms of the positive societal impact, we apply `DISCOD` to reduce the bias; bias mitigation is a crucial problem in deploying AI, preventing social issues, and improving fairness.

## REPRODUCIBILITY STATEMENT

In the appendix, we provide the implementation details, such as the model used and the hyperparameters; we provide the additional mathematical formulation. We curate the real pairs from Unsplash and will release the dataset as a list of URLs. In addition, we give an explanation of how to curate a synthetic dataset in the appendix. We use the DreamBooth dataset, Waterbirds, and CelebA for experiments.

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

# A DISCOD: DIstinct and Shared COncept Discovery

## A.1 First stage

The mutual information satisfies the following equation, $I(\mathbf{x} \mid \mathbf{y}) = H(\mathbf{x}) - H(\mathbf{x} \mid \mathbf{y})$ where $H$ is the entropy. Since we optimize $\mathbf{y}$, $H(\mathbf{x} \mid \mathbf{y})$ is related to the optimization goal. Thus, we can maximize the mutual information by minimizing the conditional entropy, $H(\mathbf{x} \mid \mathbf{y})$. The minimum of entropy is 0, and we can achieve it by making the probability 1. We maximize the conditional probability. The probability is proportional to the cosine similarity, $p(\mathbf{x} \mid \mathbf{y}) \sim \texttt{CosSim}(\texttt{CLIP}_I(\mathbf{x}), \texttt{CLIP}_T(\mathbf{y})) = \frac{\texttt{CLIP}_I(\mathbf{x}) \cdot \texttt{CLIP}_T(\mathbf{y})}{|\texttt{CLIP}_I(\mathbf{x})||\texttt{CLIP}_T(\mathbf{y})|}$ in CLIP models where $\texttt{CLIP}_I, \texttt{CLIP}_T$ are image and text encoders. The one of objective term $\hat{\mathcal{L}}_s$, the approximation of $\mathcal{L}_s$, is the following:

$$\hat{\mathcal{L}}_s(A, B, \mathbf{y}_{A \setminus B}, \mathbf{y}_{B \setminus A}, \mathbf{y}_{A \cap B}) = - \left[ I\left(A \mid \mathbf{y}_{A \setminus B}, \mathbf{y}_{A \cap B}\right) + I\left(B \mid \mathbf{y}_{B \setminus A}, \mathbf{y}_{A \cap B}\right) \right] \quad (6)$$

$$\approx \left(1 - \texttt{CosSim}\left(\texttt{CLIP}_I(A), \texttt{CLIP}_T\left([\mathbf{y}_{A \cap B}; \mathbf{y}_{A \setminus B}]\right)\right)\right) +$$

$$\left(1 - \texttt{CosSim}\left(\texttt{CLIP}_I(B), \texttt{CLIP}_T\left([\mathbf{y}_{A \cap B}; \mathbf{y}_{B \setminus A}]\right)\right)\right). \quad (7)$$

Further, we can express the mutual information with Kullback-Leibler divergence:

$$I(\mathbf{x} \mid \mathbf{y}) = \mathbb{E}\left[ D_{KL}\left(p(\mathbf{y} \mid \mathbf{x}) \| p(\mathbf{y})\right) \right], \quad (8)$$

where $D_{KL}$ is Kullback-Leibler divergence. We do not know $p(\mathbf{y})$, but we want to infuse the information bottleneck into $p(\mathbf{y} \mid \mathbf{x})$ not to contain unnecessary information; we set $p(\mathbf{y})$ as uniform distribution. As mentioned in the main paper, we compute $p(\mathbf{y} \mid \mathbf{x})$ by computing cosine similarity between the pre-trained embedding of tokens and applying the softmax operation. The another term $\hat{\mathcal{L}}_m$ is the following:

$$\hat{\mathcal{L}}_m(\mathcal{E}, \mathbf{y}_{A \setminus B}, \mathbf{y}_{B \setminus A}) = \mathbb{E}\left[ D_{KL}\left(p\left(\mathbf{y}_{A \setminus B} \mid A\right) \| U\right) \right] + \mathbb{E}\left[ D_{KL}\left(p\left(\mathbf{y}_{B \setminus A} \mid B\right) \| U\right) \right] \quad (9)$$

$$= \texttt{CE}\left(P\left(\texttt{CosSim}\left(\mathcal{E}, \mathbf{y}_{A \setminus B}\right)\right), U\right) + \texttt{CE}\left(P\left(\texttt{CosSim}\left(\mathcal{E}, \mathbf{y}_{B \setminus A}\right)\right), U\right), \quad (10)$$

where $P(\cdot)$ denotes the softmax function, $U$ is the uniform distribution, and $\mathcal{E}$ is the pre-trained token embeddings. The final objective contains the projection on discrete embedding. Since the pre-trained text tokens embed vast prior knowledge of human interpretable language, this implicitly acts as a prior. Our objective is as follows:

$$\min_{\mathbf{y}_{A \setminus B}, \mathbf{y}_{B \setminus A}, \mathbf{y}_{A \cap B} \in \mathcal{E}} \hat{\mathcal{L}}_s(A, B, \mathbf{y}_{A \setminus B}, \mathbf{y}_{B \setminus A}, \mathbf{y}_{A \cap B}) + \lambda_m \hat{\mathcal{L}}_m(\mathcal{E}, \mathbf{y}_{A \setminus B}, \mathbf{y}_{B \setminus A}) \quad (11)$$

Since we optimize the above objective on discrete embedding, we adopt PEZ method (Wen et al., 2024).

## A.2 Second stage

We use the diffusion model in the second stage. The diffusion model formulates the Gaussian distribution with time schedule as $p(\mathbf{x}) \sim \mathcal{N}(\alpha_t \mathbf{x}, \boldsymbol{\sigma}_t^2)$. Since the Kullback-Leibler divergence between two Gaussian distributions has the closed form solution, the maximization of mutual information is naturally described as the minimization of L2 loss, $\mathcal{L}_d(A, \mathbf{y}) = \|\boldsymbol{\epsilon} - \boldsymbol{\epsilon}_\theta(\mathbf{x}, t, \mathbf{y})\|_2^2$ where $\mathbf{x} \in A$. Thus, the proposed objective is the following:

$$\hat{\mathcal{L}}_s = \mathcal{L}_d(A, [\mathbf{y}_{A \cap B}; \mathbf{y}_{A \setminus B}]) + \mathcal{L}_d(B, [\mathbf{y}_{A \cap B}; \mathbf{y}_{B \setminus A}])$$

$$+ \mathcal{L}_d(\overline{A \setminus B}, \mathbf{y}_{A \setminus B}) + \mathcal{L}_d(\overline{B \setminus A}, \mathbf{y}_{B \setminus A}) + \mathcal{L}_d(\overline{B \cap A}, \mathbf{y}_{A \cap B}) \quad (12)$$

where $A$ and $B$ are the given objects, and $\overline{A \setminus B}, \overline{B \setminus A}$, and $\overline{B \cap A}$ are the synthetic images from the first stage. Unlike the first stage, we discard the information-regularized term because we use synthetic images. Our first stage and second stage maximize the information of two objects.

# B Experiments

## B.1 Implementation details

**Main experiments.** We use the two types of vision language models. The first one is CLIP (Radford et al., 2021). We use ViT-H-14 architecture; we adopt the ViT-bigG model for CLIPScore not to

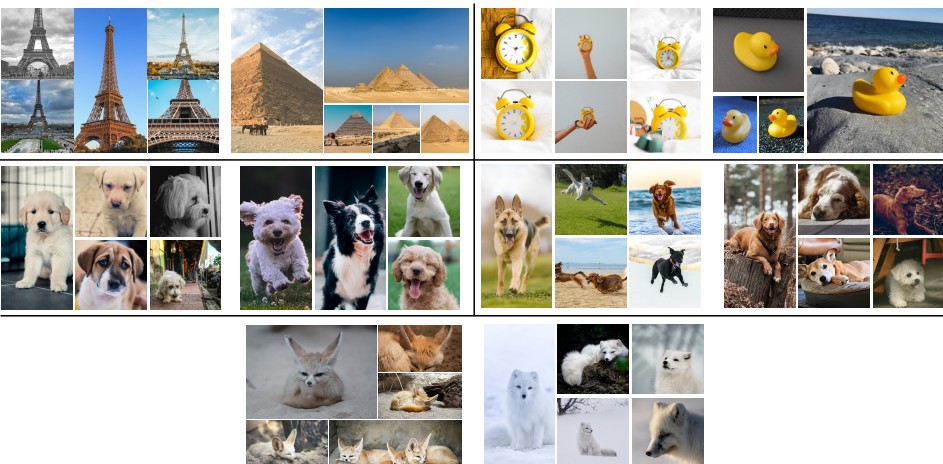

Figure 11: **Sample pairs of Real data.** Real pairs are curated from Unsplash and the DreamBooth dataset.

overlap with the used model for discovery. The second type of vision language model is latent stable diffusion (Rombach et al., 2022). We use Stable Diffusion 2.1 Base as the text-to-image (T2I) diffusion model.

In TI baseline, Vinker et al. (2023) have introduced a method for decomposing an individual instance into sub-concepts using a binary tree structure of text tokens. We adapt the method to our setting by sharing one sub-concept between two image sets in a binary tree structure. In UCD, we set the weight combination coefficient to 0.5 for each concept token.

We have hyperparameters of batch size, learning rate, and loss weight coefficient $\lambda_m$. We set the batch size of each image set as 1. Thus, the total batch size is 2. The learning rate for the baselines is based on the original, considering different batch sizes; we scale the learning rate if it improves the quality. The learning rate for DISCOD is determined in {0.1, 0.01, 0.01} in the first stage; the learning rate in the second stage is 5e-4. The coefficient of $\lambda_m$ is set as 0.1. The number of iterations is 1,000 and 100 for the first and second stages, respectively. We use a single 80G A100 for all experiments, but, the required memory consumption is much lower than 80G.

As shown in Fig. 11, We curate the real data from Unsplash to construct the real datasets; one pair is taken from the DreamBooth dataset. For synthetic data, we generate the images from Stable Diffusion 2.1 Base by text. For our task, we need a pair of two objects. Thus, we overlap one concept between pairs and provide distinct concepts for each object. We generate the pairs on the pre-defined text template and concepts. We use the following concepts:

- Category:     `sedan, bus, motorcycle, ship, airplane, truck, train, shirt, pants, shoes, dress, cap, chair, table, bench,`

- Color: `red, yellow, green, purple, blue`

The category or color can be overlapped between two objects. The total number of pairs is 30, and each pair has 10 images. Figure 12 shows the samples of pairs.

**Disentanglement in Text-To-Image Personalization.** We first discover the shared and distinctive concepts by applying DISCOD in the first stage. We fine-tune the diffusion model by providing the specific prompt, *e.g.*, "the photo of $\mathbf{y}_{A \cap B}\ \mathbf{y}_{A \setminus B}$ object", rather than "the photo of object". In other words, we replace the textual inversion in the second stage with the fine-tuning of T2I diffusion models. For this experiment, we use Stable Diffusion XL and LoRA. The batch size is 1, the learning rate is 1e-4, and the number of iterations is 500 for fine-tuning. The rank of LoRA is 4.

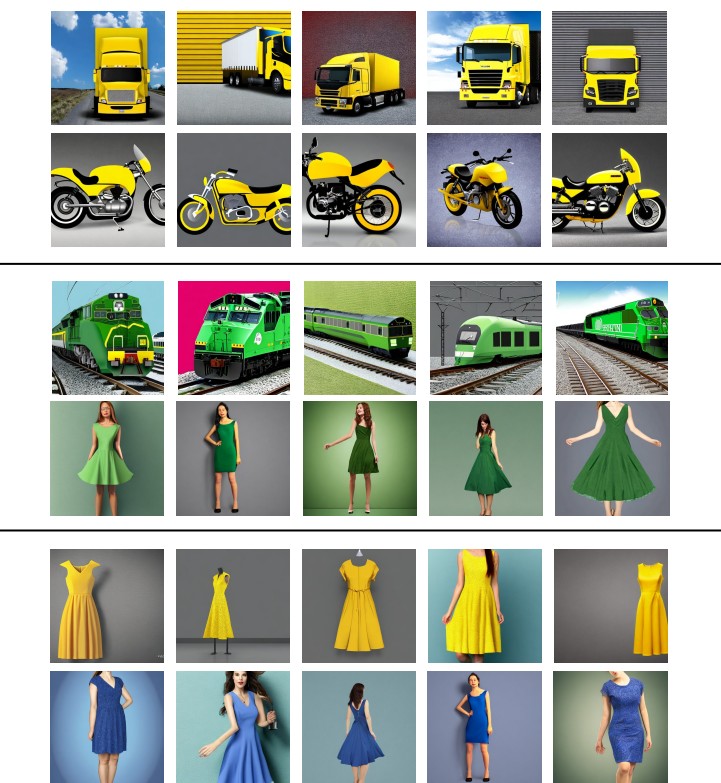

Figure 12: **Sample pairs of Synthetic data.** Synthetic pairs are generated from Stable Diffusion 2.1 Base with the pre-defined category and color. In the pair, one concept is overlapped, and the other concepts are not overlapped; both category and color can be a commonality.

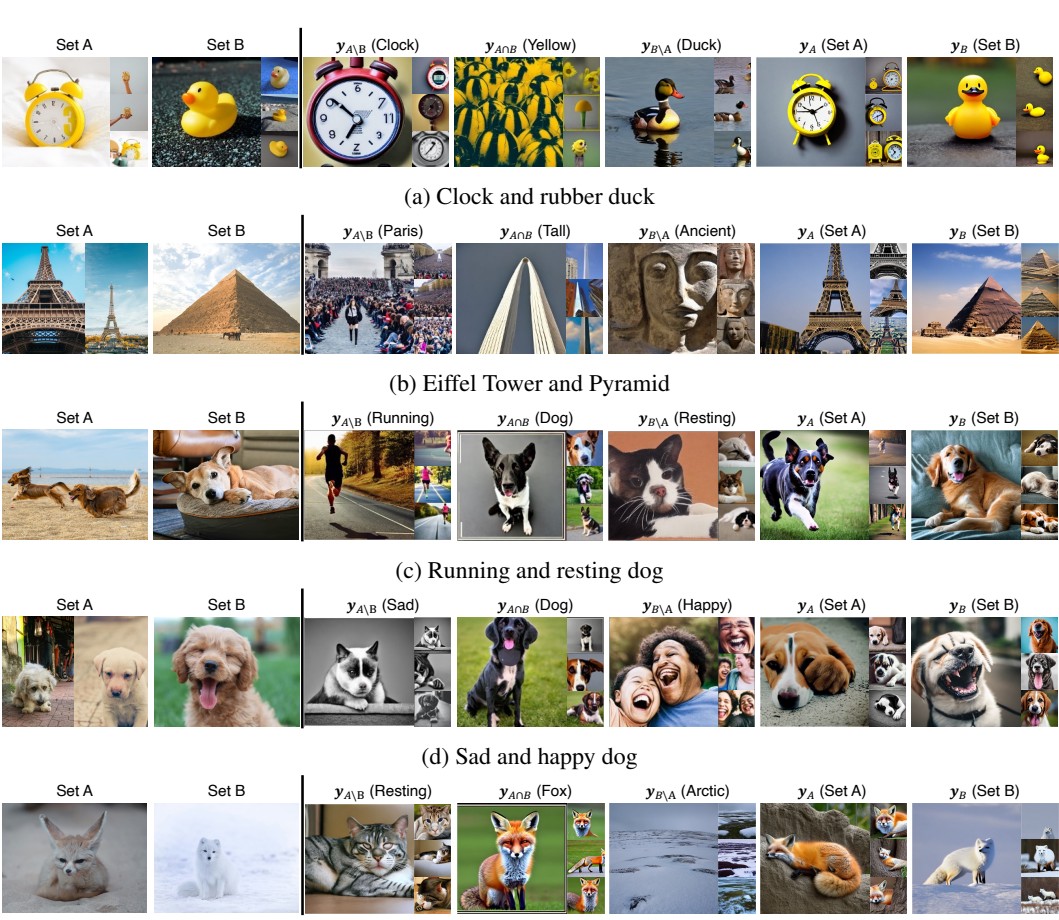

Figure 13: **Qualitative and composite results on real pairs.** We apply DISCOD to the real pairs from Set 1 and Set 2. The discovered concepts denoted as $\mathbf{y}_{A \setminus B}, \mathbf{y}_{B \setminus A}, \mathbf{y}_{A \cap B}$ are generated and visualized with the concepts indicated above. We also present the composite results, $\mathbf{y}_{A \setminus B} + \mathbf{y}_{A \cap B}$ and $\mathbf{y}_{B \setminus A} + \mathbf{y}_{A \cap B}$, where $+$ is the concatenation operation. We simply denote it as $\mathbf{y}_A, \mathbf{y}_B$.

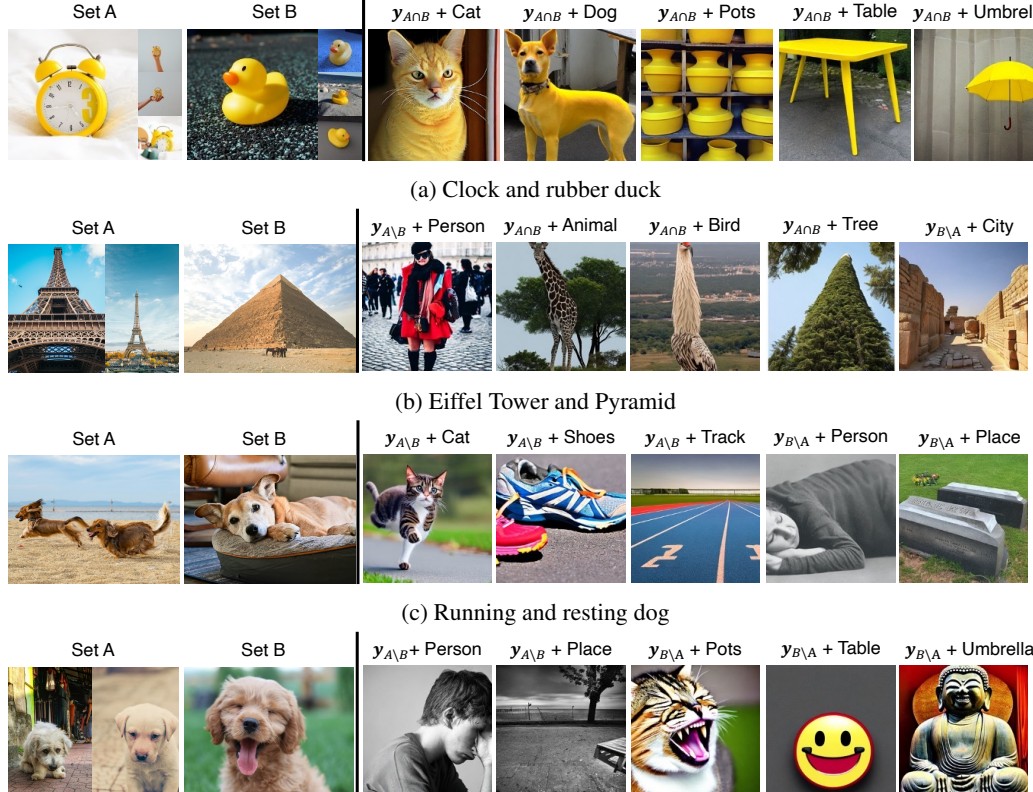

(a) Clock and rubber duck

(b) Eiffel Tower and Pyramid

(c) Running and resting dog

(d) Sad and happy dog

Figure 14: **Discovered concept with additional text token.** We generate the discovered concepts with arbitrary text tokens. It help us understand the meaning of discovered concepts.

### B.2 QUALITATIVE RESULTS

**Discovered concepts.** In the main paper, we only show the discovered concepts of $\mathbf{y}_{A \setminus B}, \mathbf{y}_{B \setminus A}$, and $\mathbf{y}_{A \cap B}$. We present the composite results of $\mathbf{y}_{A \setminus B} + \mathbf{y}_{A \cap B}$ and $\mathbf{y}_{B \setminus A} + \mathbf{y}_{B \setminus A}$ as shown in Fig. 13. We observe that the composite results are reasonable to the given two set. For example, as shown in Fig. 13b, $\mathbf{y}_{A \setminus B} + \mathbf{y}_{A \cap B}$ and $\mathbf{y}_{B \setminus A} + \mathbf{y}_{B \setminus A}$ represent the Eiffel Tower and Pyramid.

**Discovered concepts with additional text.** We visualize more examples of the discovered concepts with the arbitrary text as shown in Fig. 14. It validates the meaning of the discovered concepts, $\mathbf{y}_{A \setminus B}, \mathbf{y}_{B \setminus A}$, and $\mathbf{y}_{A \cap B}$. For example, in Fig. 14b, $\mathbf{y}_{A \cap B}$ with animal, bird, and tree generates the corresponding objects with the tall notion.

**Human study of alignment and aggrement.** Alignment measures whether the written answer of participants is the same as the concept of generated images, and agreement measures the amount that the discovered concepts are reasonable. In other words, the alignment is judged without showing the discovered concepts, and the agreement is computed after showing the discovered concepts. As shown in Fig. 15, the agreement is improved. The alignment is high when the concepts are easily identified like sad or happy as shown in Fig. 15a

The example shown in Fig. 15c seems difficult from the result in Fig. 15d. The scores of commonality are lower than others. Figure 16 shows the short answer written by participants. Most of the written answers from participants are about architecture or landmarks; thus, the alignment is low. However, The fifth most common response is related to height. The agreement is better than alignment, although the generated images are not straightforward.

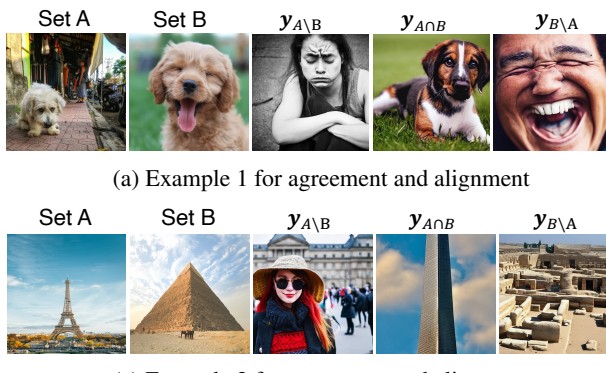

| | Set A | Set B | $\boldsymbol{y}_{A \backslash B}$ | $\boldsymbol{y}_{A \cap B}$ | $\boldsymbol{y}_{B \backslash A}$ |

| Concept | Alignment | Aggrement |
|---|---|---|
| $\mathbf{y}_{A \backslash B}$ | 0.86 | 0.94 |
| $\mathbf{y}_{A \cap B}$ | 1.0 | 1.0 |
| $\mathbf{y}_{B \backslash A}$ | 0.86 | 1.0 |

(a) Example 1 for agreement and alignment

(b) Result of Example 1

| Concept | Alignment | Aggrement |
|---|---|---|
| $\mathbf{y}_{A \backslash B}$ | 0.48 | 0.74 |
| $\mathbf{y}_{A \cap B}$ | 0.19 | 0.50 |
| $\mathbf{y}_{B \backslash A}$ | 0.86 | 0.94 |

(c) Example 2 for agreement and alignment

(d) Result of Example 2

Figure 15: **Human study of alignment and aggrement.** We conduct a human study about alignment and agreement. Alignment measures whether the short answer, written by participants, exactly matches with the discovered concepts from DISCOD. Aggrement measures whether the discovered concepts are reasonable and recognizable.

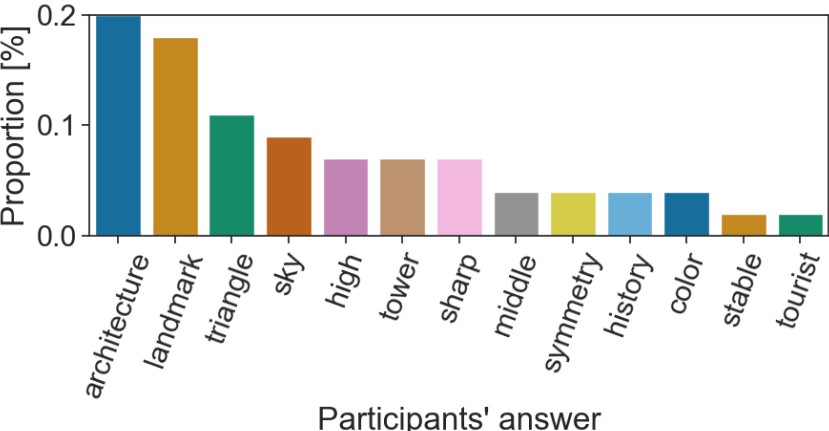

Figure 16: **Participants' short answer of commonality in Eiffel tower & pyramid.** The architecture and landmark show a high proportion. The high concept is also shown in the short answer. Thus, the agreement is improved compared to the alignment.

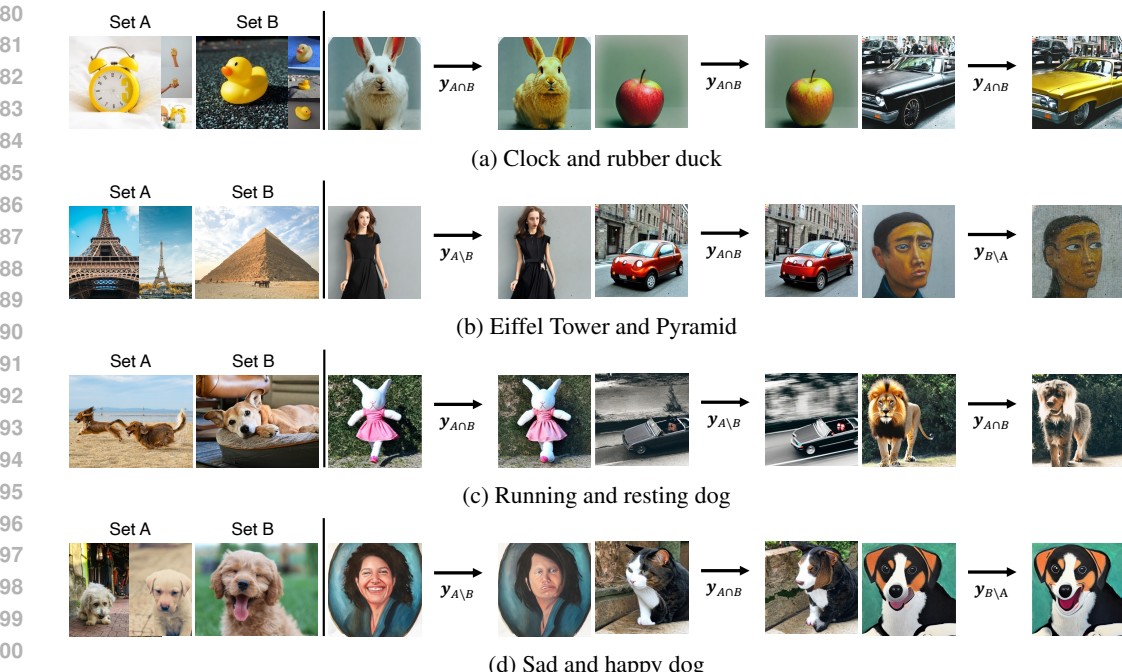

Figure 17: **Image editing with discovered concept.** We combine Prompt-to-Prompt with the discovered concepts by `DISCOD`. We can see that the discovered concepts can edit the given images with their meaning.

## B.3 IMAGE EDITING

We visualize more examples of the discovered concepts with the editing algorithm. Figure 17 shows the several editing results. We can notice that the editing reflects the discovered concepts, respectively.

## B.4 DISENTANGLEMENT IN TEXT-TO-IMAGE PERSONALIZATION

We present more examples with the same prompts to validate that DreamBooth with `DISCOD` mitigates the bias problems with high chances. The randomly generated examples can validate that `DISCOD` resolves the entanglement of the undesirable attributes with a high chance. As shown in Fig. 18, `DISCOD` can mitigate the biases issues in personalization.

## B.5 BIAS EXPERIMENTS

**Mitigating biases in classification.** The Waterbirds dataset (Sagawa et al., 2020) is composed of the bird photographs from CUB dataset (birds) and Places dataset (background). The classes are waterbird or landbird, and the places are water background and land background. To control bias, they construct the dataset as 5% of waterbird on the land background and 5% of landbird on the water background. The waterbirds on land are the smallest group. CelebA is the face dataset with the hair color of blond or dark with gender bias of male and female. The blond-haired males are the smallest group.

**Zero-shot prompting.** We use the 80-prompts of which an example is "`the photo of a {class}`." For `{class}`, we adopt {landbird, waterbird} for Waterbirds dataset and {blond, non-blond} for CelebA. After extracting the text features of 80-prompts, we take an ensemble of these features: (1) normalize the features, (2) average the features, and (3) normalize the feature again. It does not require additional training.

**Group zero-shot prompting.** If we know the group information causing the bias or spurious correlation, we can enhance the prompting template used in zero-shot prompting by infusing the

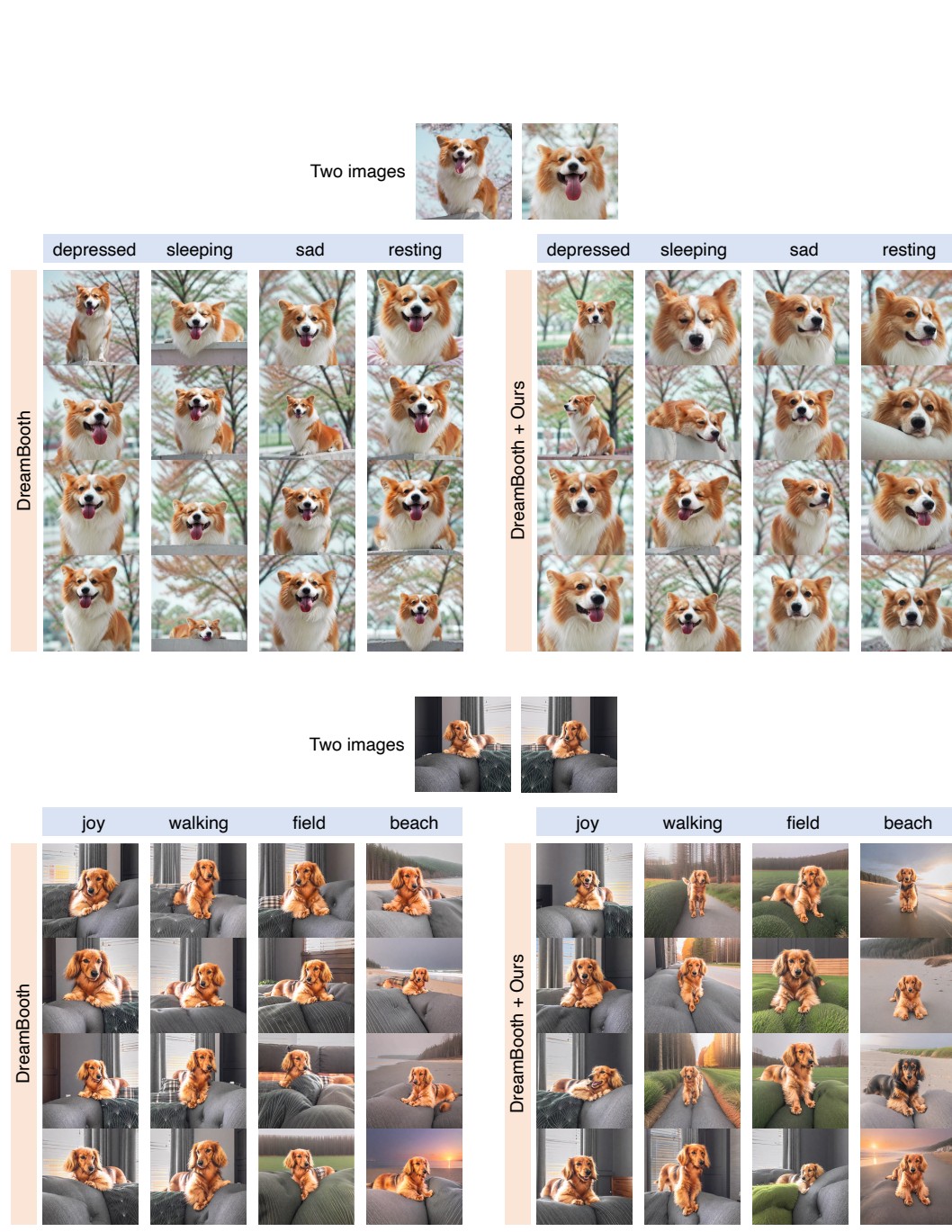

Figure 18: **Disentanglement in DreamBooth with and without** `DISCOD`**.** We provide more generated examples to validate the effectiveness of `DISCOD`. We observe that `DISCOD` can mitigate the bias problems.

biased information into the templates. In Waterbirds dataset, the example of the used template is "`the photo of a {class} on a {group}.`" The group can be {water background, land background}; In CelebA dataset, the example of the used template is "`the photo of a male of {class}.`" Since the well-known gender bias in the dataset is male, we only use male information. Note the performance of using only male information is better than the one of using female or both information.

**B2T.** B2T discovers the bias from mispredicted images without the knowledge of the bias information. We use the prompt design reported in the original paper. For Waterbirds, the used keywords are `forest, woods, tree, branch, ocean, beach, lake, surfer, water, boat, dock, rocks, sunset, kite, sky, flight, flies`; for CelebA, the used keywords are `man, player, person, artist, comedy, film, actor, face`.

**Ours.** Like B2T, we apply `DISCOD` to the mispredicted classes; waterbirds and landbird classes in Waterbirds dataset, and non-blond and blond classes in CelebA dataset. When we discover the shared and distinct concepts, we use the templates like "`a photo of {$\mathbf{y}_{A \cap B}$} {with, in, of} a {$\mathbf{y}_{A \backslash B}$ or $\mathbf{y}_{B \backslash A}$}`". After discovering these concepts, we infuse $\mathbf{y}_{A \backslash B}$ and $\mathbf{y}_{B \backslash A}$ into the 80 prompts. Note that B2T and `DISCOD` do not use the bias information by humans.

As visualized in the main paper, the discovered concepts in Waterbird are the water and land backgrounds, and the discovered concepts in CelebA are the male and female. Thus, the prompting with biased information improves the worst-case accuracy. We can also observe that the improvement is effective compared to other baselines. We hypothesize that our discovered biases are what VLMs recognize between mispredicted sets. Thus, we help VLMs improve their recognition ability more.

