# OpenReview forum: "Distinct and Shared Concept Discovery for Fine-grained Concept Inversion"
_ICLR.cc/2025/Conference — ICLR 2025 Conference Withdrawn Submission_

### Official Review · Reviewer_2ucC · 2024-10-18

**Soundness:** 2
**Presentation:** 4
**Contribution:** 2
**Rating:** 5
**Confidence:** 4

**Summary:**

This paper investigates the task of concept discovery and inversion. The authors introduce a novel approach that extends beyond the identification of shared concepts, as explored in prior work, to also discover distinct concepts between two image sets. They propose a two-stage framework to both uncover and refine these concepts, supported by well-designed objective functions.

The experimental results highlight the improved performance of the proposed method in concept discovery, inversion, and composition, while also demonstrating additional capabilities.

Moreover, the paper explores three compelling applications enabled by this method, effectively showcasing its practical utility.

**Strengths:**

- The writing, structure, and presentation of the manuscript are clear and enjoyable. The text is easy to read, and the experiments are well explained. The layout design is also commendable.
- The motivation is well-articulated and valid: "Further discover distinct concepts." This idea is novel and adds value to the field.
- The proposed framework and objective functions are technically sound, with the decomposed mutual information terms being particularly well-designed.
- The experiments are diverse and thoughtfully planned, yielding good results.
- The application study is interesting and practical. I believe concept discovery holds significant practical value. However, the implementation behind the application results is somewhat unclear (see further comments below).

**Weaknesses:**

**General major concerns**

(1) Lack of comparison: The most related method, MCPL [1], which can achieve similar functionality, is not compared in the experiments and not discussed in the related work.

(2) In the main experiments and appendix, the qualitative results are limited to a few concepts. It is necessary to show more qualitative results in the appendix to demonstrate the effectiveness of the proposed method.

(3) Further discussion with closely related concept (category) discovery works, such as [2] and [3], is needed.

**Major concerns about the application studies**

**(A) Re@Image Editing application**
1. As shown in Figure 8, how is the semantic meaning (e.g., "Ancient") of the discovered concepts obtained? Are these semantic meanings annotated by humans?

2. For this application study, it is necessary to compare with prior concept discovery methods such as Text Inversion or MCPL.

**(B) Re@Disentanglement in Text-to-Image Personalization**
1. Given a few images of the target instance for diffusion model personalization, how are Set A and Set B defined to discover shared and distinctive concepts?

2. How are these diverse concepts, such as "depressed," "sleeping," "sad," or "resting," discovered and obtained?

3. From the results shown in Figure 9 (**Left**), even with the proposed method, the backgrounds of the generated dog images are clearly biased towards the few input images of the target instance. Why is this bias mitigated?

**(C) Re@Mitigating Group Bias**
1. The implementation description for this application study is not clear.

2. My major question about this application study is: How are the biases discovered?
```
Let’s assume Set 1 contains mis-predicted images and Set 2 contains correctly predicted images. Set 1 will have a significant number of concepts that differ from Set 2 — i.e., CelebA images have **5 landmark locations, 40 binary attribute annotations per image**, leading to a large, diverse set of concepts present in Set 1 but not in Set 2.
```

Then, my questions are:

i) How is the semantic meaning (concept names such as water, land, male, female, etc.) for these discovered concepts obtained? Manually name the concetps?

ii) Among all the discovered concepts, how are the valid biases—**Background (habitat) in Waterbird and Blond-hair Male in CelebA**—identified? Manually select from all the concepts? This is a critical concern. As shown in [2], the same dataset can have biases along many dimensions (such as the 5 types of locations and 40 binary attributes).

Biases should be quantified. If we do not quantify it, how can we know it is a bias?

**Minor concerns**

(1) Why is this work described as "FINE-GRAINED" concept inversion, as stated in the title? How is "fine-grained" defined in this context?

(2) One major claim repeatedly stressed is: "Prior object discovery works focus on extracting commonalities and are not capable of identifying differences" (P1#L50), which makes them "...discovered textual tokens can be adversarial; certain textual tokens can encode a large coverage of information, making other textual token concepts meaningless…" (P3#L120). → What are the evidences, preliminary experimental results, or proof to support this claim? If a claim is made, it should be verified by either prior works or the current work.

(3) In the current writing and presentation, it is unclear what a "concept" is. It is recommended to clarify and illustrate this from the beginning of the work so that readers can carry a clear understanding of "concept" throughout the rest of the paper.


**References**

[1] Jin, C., Tanno, R., Saseendran, A., Diethe, T., & Teare, P. (2023). An image is worth multiple words: Learning object level concepts using multi-concept prompt learning. In ICML, 2024.

[2] Yang, Y., Zhang, H., Katabi, D., & Ghassemi, M. (2023). Change is hard: A closer look at subpopulation shift. In ICML, 2023.

[2] Han, K., Li, Y., Vaze, S., Li, J., & Jia, X. (2023). What's in a Name? Beyond Class Indices for Image Recognition. In CVPRW, 2024

[3] Liu, M., Roy, S., Li, W., Zhong, Z., Sebe, N., & Ricci, E. (2024). Democratizing fine-grained visual recognition with large language models. In ICLR, 2024.

**Questions:**

(1) In the context of this work, "concept" is defined as object attributes or components. My question is: to discover these "concepts" from an image set, why do we need to recognize them at the "concept" level from complex object composition in real-world scenes and their entanglement with various other concepts? Can’t we discover class names and simply source (Wikipedia) / generate (LLM) all the possible concepts related to the class?

(2) What does the linked URL in the footnote at P1#L38 mean? Why is it related to "the creation of novel objects"?

(3) Again, how is a concept defined? Taking the example illustrated at P3#L135:
Let:
$A$ ‎ =  "yellow chair", and $B$‎ = "yellow table".

Then, following the authors' claim:

The ideal solution is:

$y_{A \setminus B}$ = "chair",  $y_{B \setminus A}$ = "table", and $y_{A \cap B}$ = "yellow".

And a failure solution is:

$y_{A \setminus B}$  = "yellow chair",  $y_{B \setminus A}$ = "yellow table", and $y_{A \cap B}$ = "photo".

Is this failure solution wrong? In my opinion, this is not wrong. It is a matter of semantic granularity, or concept granularity. If we consider "yellow chair" and "yellow table" as a whole, the failure solution is not wrong.

On the other hand, how about "climbing gym" and "climbing club"? In addition to granularity, there is also semantic ambiguity between concepts. Here, "gym" and "club" refer to the same concept.

How are the concerns above handled? This ambiguity puzzles me.

(4) Why is optimizing Eq. 2 difficult? This is not described.

(5) What will be the discovery results in the cases where: i) Set A and Set B have totally identical concepts?; ii) Set A and Set B have totally disjoint concepts? And what will the generated images be?

---

### Official Review · Reviewer_HTRr · 2024-10-31

**Soundness:** 2
**Presentation:** 1
**Contribution:** 2
**Rating:** 3
**Confidence:** 4

**Summary:**

The authors introduce a novel task to identify both shared and unique concepts between two sets of images. They propose a two-stage textual inversion method to extract these concepts effectively. Stage 1 maps the concepts to existing text tokens in CLIP text encoder via discrete parameterization and optimization. Stage 2 refines these text token embeddings via latent diffusion loss.

**Strengths:**

1. The problem setting is new. Analyzing concept overlap and distinction between two sets of images hasn't been thoroughly explored in previous works.

2. The learning objective in stage 1 is simple and intuitive.

3. The authors showcased some scenarios where deriving shared and distinct concepts of two sets of images may be helpful.

**Weaknesses:**

1. Bad writing and poor readability.

    a. The authors did not clarify their main contributions, burdening the reviewers to compare their work with relevant works in detail.

    b. The diagrams are hard to understand. The poster frame (Figure 1) barely contains any information about the network architecture, training objective, or the form of representation, and the arrows don't make any sense. The experiment result diagrams are also confusing. See the question section for particular concerns.

2. Insufficient experiment.

    a. There are no quantitative results that proves the effectiveness of stage two. There are 5 terms in the loss function of stage 2, including terms that encourage reconstruction and terms that encourages concept discovery. The authors should do a more detailed analysis instead of just showing us figure 6. Figure 6 tells us nothing about how stage 2 helps with the concept discrimination except that it overfits the original image, which directly contradicts with the motivation of information regularization. Furthermore, stage 2 isn't even applied in section 3.4, as they only use the discoverd concept in stage 1 as group labels.

    b. The authors often try to prove some points with only one sample, as in figure 3, figure 5b, figure 6, figure 7, and table 2. This greatly undermines the soundness of the technical points of the paper.

3. Mathematical deductions are lengthy and unnecessary. The content regarding mutual information is neither an original theorectical contribution nor a sufficient guarantee for the soundness of the method. For instance, approximating conditional probability via cosine similarity scores, though common in representation learning, is not exact and introduces an (unbounded) approximation error because cosine similarity is not strictly a probability measure. The effectiveness of this approximation mainly relies on careful downstream tuning and optimization. Despite the seemingly rigorous math, the proposed method is mostly an intuition-based, empirical and heuristic framework. It's better for the authors to simply point out the intuitives behind ther designs and back their method with more sufficient experiment analysis instead of wasting long paragraphs on math.

**Questions:**

1. Figure 5b: what does the TI and UCD labels mean when you're showing the effect of regularization loss?

2. Figure 10: what kind of bias is this figure trying to show? There are only a few pictures and I don't see any obvious pattern or bias.

3. The proposed method requires two sets of images to discover the concepts. In the each of the experiments in section 3.1-3.4, what are the two image sets that you learn from, respectively? Do you learn the concept for each image respectively?

5. Human studies: how many participants are involved and how many instances are each of them required to evaluate?

---

### Official Review · Reviewer_UT4B · 2024-11-01

**Soundness:** 3
**Presentation:** 2
**Contribution:** 3
**Rating:** 6
**Confidence:** 3

**Summary:**

This paper introduces a two-stage framework for distinct and shared concept discovery (DISCOD). Instead of focusing solely on identifying shared concepts within images of individual identities, DISCOD identifies shared concepts across multiple identities while preserving the unique concepts inherent to each. In the first stage, DISCOD employs information-regularized textual inversion, aiming to separate representative concepts that are distinctive from others while capturing shared concepts among different objects. In the second stage, it optimizes these representations to align the composite concepts with their corresponding objects. The authors evaluate DISCOD’s performance on various tasks, including image editing, text-to-image personalization, and group bias mitigation.

**Strengths:**

1. This paper addresses a new problem of discovering both shared and distinct concepts within and across two groups of objects.

2. The proposed information-regularized textual inversion method identifies distinct concepts while maximizing the separation of shared concepts, enabling recognition of both commonalities and differences among concepts. It is well designed and explained.

3. Experimental results on three tasks demonstrate good performance.

**Weaknesses:**

1. Some parts of the manuscript are unclear. For example, the object concepts A\B, B\A and A∩B are represented with discrete embeddings. It is not clearly described how these embeddings are used in the diffusion model for generating images in the second stage, especially in Figure 1 and Section 2.3. Because they are not text prompts.

2. Another main concern is the preparation of training pairs (i.e., A-B pairs). Since A and B can represent any concept in the dataset, and object sets may vary in the number of concept categories, how does the method address potential pair imbalance issues?

3. The demos in the last column of Figure 2 could benefit from more consistent tests, such as showing the results of Y_A\B, Y_B\A, Y_A∩B + new concepts.

**Questions:**

See the weaknesses.

---

### Official Review · Reviewer_hDza · 2024-11-04

**Soundness:** 3
**Presentation:** 2
**Contribution:** 3
**Rating:** 6
**Confidence:** 3

**Summary:**

This manuscript introduces DISCOD, a framework designed to simultaneously discover both shared and distinct concepts between different sets of images. By effectively separating unique characteristics from common properties among objects, DISCOD enhances the understanding and manipulation of visual concepts. The method demonstrates improved performance in concept discovery over baseline approaches, as evidenced by quantitative metrics like CLIPScore, success rate, and validation through human studies.

**Strengths:**

1. The manuscript presents a clear and well-articulated motivation for separating shared and distinct concepts among objects.
2. It provides a substantial amount of experimental results that effectively demonstrate the validity of their proposed method. The numerous experiments are impressive and give strong support to their idea of separating shared and distinct concepts among objects using textual inversion.

**Weaknesses:**

1. This manuscript tries to separate shared and distinct concepts among objects. Even though the visuals and downstream tasks show it can achieve concept separation through textual inversion, it's hard to see why previous concept-related methods can't do this too. It would be better to explain this in the motivation section instead of just in the pipeline and the visualizations in the downstream tasks.
2. While the paper introduces the DISCOD framework for separating shared and distinct concepts using textual inversion, the technical contributions seem somewhat limited. The authors incorporate regularization methods to distill concepts from images to text and further optimize text representations by fine-tuning using generated synthetic images. Although these techniques enhance the existing methods, I hoped for more substantial insights from the paper.

**Questions:**

1. Figure 1 isn't informative enough; it doesn't convey the main idea effectively.
2. Why use a uniform distribution for the KL divergence in information regularization? Can other distance measurement methods be used instead?
3. The attention maps are hard to interpret. Concepts like "tall" are abstract and difficult to visualize, making it tough to see the commonalities and differences.
4. Figure 5(b) shows the performance of TI and UCD but not DISCOD. Why isn't DISCOD included? Is it a mistake?
5. Interestingly, in Figure 18, the dog's color changes at the end of the third row. Why does this happen?

---

### Note · Authors · 2024-11-14

**Comment:**

We greatly appreciate the reviewers' time, effort, and constructive feedback. We have decided to withdraw our paper.

**Withdrawal Confirmation:**

I have read and agree with the venue's withdrawal policy on behalf of myself and my co-authors.